# To participate or not to participate? A qualitative investigation of students' complex motivations for verbal classroom participation

**Emilee Severe** [ID]**, Jack Stalnaker, Anika Hubbard, Courtni H. Hafen, Elizabeth G. Bailey** [ID]*

Department of Biology, Brigham Young University, Provo, Utah, United States of America

* liz_bailey@byu.edu

**Data Availability Statement:** Our dataset includes the transcripts of individuals interviews. Rather than providing the full transcripts, we share frequent quotes from the interviews throughout the

## Abstract

Previous research has suggested that making classrooms more active and student centered improves learning, and this usually involves encouraging student talk in the classroom. However, the majority of students remain silent during whole-class discussions, and men's voices are more likely to be heard in science classrooms. Previous interview studies and quantitative studies have discussed the role instructors play in encouraging or discouraging participation, the weight students put into the fear of negative evaluation, and other factors. However, interview studies on the experiences of college students in the sciences, specifically, are lacking. Thus, we conducted a qualitative interview study to investigate students' experiences deciding whether to participate verbally in class, focusing on students recruited from science classrooms. We analyzed the data using an inductive approach and found three main themes: (1) A wide variety of external factors impact students' decision to participate, including instructor characteristics and choices, peer influences, and course material characteristics; (2) Students weigh these factors in complex ways, and this internal calculus varies by student; and (3) Women put greater emphasis on fearing peer judgment, and men may be more motivated by course material considerations. Most of the external factors we identified as important for student participation have been described previously, and we validate that previous literature. We add to the literature by a more complex discussion of how students weigh these different factors and how complex the classroom ecosystem can be. We end by framing our results within the Expectancy Value Theory of motivation, discussing limitations, and providing implications for science college instructors to promote broad and equitable participation.

## Introduction

Education research has provided more and more evidence that active, student-centered classrooms increase learning in the sciences beyond classic lecture [1–6], and incorporating "Student Talk" could be considered the common thread in most types of active, student-centered pedagogies [7]. However, while we may be confident that incorporating student talk "works" better than having students passively listen, there have been calls for further research on "what

manuscript. Including the whole transcripts could provide identifiable information about research participants, which would violate consent agreements per the Brigham Young University Institutional Review Board. Contact Sandee Aina (Sandee.Aina@byu.edu) regarding data restriction and access requests. She is the; Associate Director of the BYU Human Research Protection Program and oversees the IRB. She is the point of contact overseeing data use for this project, but she was not an author on the study.

**Funding:** The authors acknowledge internal funding from Brigham Young University's College of Life Sciences awarded to authors ES and AH as a College Undergraduate Research Award. The funders did not play a role in the study design, data collection and analysis, decision to publish, or preparation of the manuscript.

**Competing interests:** The authors have declared that no competing interests exist.

working means, for whom, and in what contexts" [8]. Regarding verbal participation in science classes, the literature suggests that the "for whom and in what contexts" questions of student talk are important, as reviewed below.

## Previous studies on who participates in science classrooms

As most instructors will anecdotally share, most students never participate in whole-class discussions in large science classrooms, and this is supported by the literature. For example, we previously found that on average, less than one-third of students participated verbally at least once when we observed university classes in the life sciences [9]. In a survey study, Nadile, Alfonso [10] reported that over half of college participants said they never ask or answer questions in large-enrollment science classes.

Of those who do choose to speak up in class, it is well documented that male voices generally dominate in classrooms, from elementary education through graduate school [11]. Multiple studies conducted in college sciences classrooms have found the same trend, with men being more likely to participate vocally in class than women on average [e.g., 9, 12, 13]. However, participation differences by gender do vary by class, and the degree to which men's voices dominate has been seen to be predicted by classroom gender ratios, class size, and competition for getting called on [9, 11, 13, 14]. Due to the small number of students who participate and the differences by gender, researchers have been interested in what motivates students to participate in class or not, as a greater understanding of students' experiences would aid in increasing participation.

## Previous studies on what motivates students to participate or not

Studies in middle schools and high schools have revealed the importance of teachers, peers, and the relevance of material. In a case study conducted in high school physics classes, both boys and girls were found to have fear of participating since they did not want to be wrong in front of their peers, but this was more common for girls [15]. In an interview study with middle schoolers and high schoolers conducted by Fredricks, Hofkens [16], they found that students participated more in student-centered science classrooms and when their peers were very engaged. While all students were motivated to participate when material was relevant, girls seemed more impacted by personal relevance while boys talked about relevance to future careers. Girls also benefited from teacher support more than male peers [16].

College interview studies on classrooms of all disciplines also highlight the importance of instructors. Auster and MacRone [17] suggested that college faculty members can encourage participation by using students' names, positively giving encouragement and approval (which is especially meaningful for women), asking analytical questions rather than factual ones (which is especially meaningful for men), providing students with enough time to respond, and asking students for their thoughts even when they do not volunteer. They also found that men were more likely to report feeling comfortable participating than women were [17]. Fassinger [18] also found that instructors were important, but they found that their pedagogical choices that allow for participation opportunities or reward participation with grade increases had a bigger impact than their personality or demeanor. In a Malaysian university, researchers found that instructor characteristics such as warmth, openness, learning names, asking probing questions, and good teaching skills could increase participation [19].

These college interview studies also highlighted factors other than the instructor, that impact students' participation. Fassinger [18] found that smaller class sizes, a positive emotional climate (especially for women), and student confidence all had a positive impact on students' participation habits. Mustapha, Abd Rahman [19] reported that when students feared

making mistakes, lacked confidence, or were not prepared for class, they were less likely to participate.

Finally, two recent quantitative survey studies were conducted to investigate students' motivations for participating in large-enrollment university science classes. They found that the majority of college students feel uncomfortable participating, and this was significantly truer for women [10, 20]. Fear of peer judgment and lack of confidence with the course material were the most common reasons students did not feel comfortable, and women were more likely to have the fear of negative evaluation than men [10, 20]. A majority of students also responded that the large number of students in the class and negative responses from the instructor to other students made them feel uncomfortable participating [20].

## Research questions

Several studies have examined motivations behind student participation in college science classrooms through observing classrooms or quantitative survey studies, but interview studies are needed to understand the complexity behind this student behavior [21]. Previous interview studies on this topic, as reviewed above, have been done in K-12 classrooms or without a focus on a specific discipline. Thus, we aimed to conduct a qualitative interview study to answer the following research questions:

1. What factors influence students' decisions to participate or not in lectures?

2. Do any of these factors seem more important for men versus women?

While we asked students about their experiences in all of their classes, we recruited students from science classrooms, specifically. Thus, our findings can increase our understanding of what prohibits college science students from speaking up in class, and thus inform ways instructors can encourage broad and equitable participation.

## Methods

### Ethics statement

Prior to data collection and analysis, we obtained approval from the Brigham Young University Institutional Review Board (Protocol E2020-069). All participants gave written consent to be interviewed and have their responses analyzed as part of this research study, and they were compensated with $20 gift cards or $20 loaded onto their student card. Participants were recruited between June 2019 and April 2020, and data were accessed for research purposes from then until the end of 2022. Researchers had access to access to information that could identify individual participants, but all ethics regulations were followed and data were kept private.

### Researchers' identities and positionality

The principal investigator (EGB) identifies as a white woman. She holds a Ph.D. and conducts research on women's and Indigenous students' experiences in science classrooms. The interviewer (ES) is a white woman and was an undergraduate student at the time of this study. This interviewer was chosen in an effort to help participants feel comfortable being open and truthful in describing their experiences in the classroom, as there would be no power differential with a peer as there would be with a professor. The analysis team was additionally composed of three women (authors AH and CHH, and one woman who only participated in group discussions about main themes) and two men (JS and one man who only participated in larger research group discussions), all white undergraduate students. The principal investigator and

the entire research team are all members of the church that sponsors the institution at which this study was conducted. Thus, the research team shared religious and cultural contexts with the research participants. This shared context, as well as a shared understanding of the institution at which these students attend class, allowed the research team to understand specific references used by the research subjects.

## Participant characteristics and recruitment process

We interviewed 19 students who had taken courses related to bioinformatics and computer science at a large, private university. Because this university is associated with a religious organization, the student body is primarily composed of religious students. Although students were recruited from science, technology, engineering, and mathematics (STEM) classes, some students were not STEM majors. Of the 19 students interviewed, nine were men, 10 were women; 15 were STEM majors, four were non-STEM majors (see Table 1). We use pseudonyms throughout the paper.

In order to increase the likelihood that a variety of perceptions would be represented, we asked professors teaching courses related to bioinformatics to identify students that fit into the following four categories: high-performers and high-participators; high-performers and low-participators; low-performers and low-participators; low-performers and high-participators. We defined "high" as in the top 20% and "low" as in the bottom 20% when asking instructors for students from these categories. We ended up with a list of 43 women and 25 men from all categories and emailed them all inviting them to be interviewed. The interviewer and research staff analyzing the interviews were blinded to the categories until after the analysis was complete so as to not bias the interview and analysis. Of the 43 women and 25 men, 10 women and 9 men responded, agreed to participate in the study, and showed up for the interview. The number of women and men in each category (high vs low performers, high vs low participators) are shown in Table 1. All categories were represented, except there were no women who were low performers and high participators. Participants generally self-identified in the same category as they were labeled by their instructors, but participants sometimes called themselves average performers while the instructors labeled them in the top 20% and often thought they participated less than instructors remembered.

Our population was relatively homogenous (all students at the same university who had taken at least one class required for the bioinformatics major), and as we approached our final number of 19 interviews, we continued to see similar themes without many new ideas. We thus concluded we had reached data saturation. This is supported by the literature. Guest, Namey [22] found that even six to seven interviews can capture the majority of themes in a homogenous population (what they defined as >80% saturation), and 11–12 interviews increased saturation to >95%. Weller, Vickers [23] discuss the idea of ensuring your sample

**Table 1. Summary of participant characteristics.**

|  | Men | Women |
|---|:---:|:---:|
| # of Participants: Total (STEM, non-STEM) | 9 (7, 2) | 10 (8, 2) |
| Age: Mean (Min-Max) | 21.7 (19–24) | 21.7 (18–25) |
| Years in College: Mean (Min-Max) | 2.4 (1–5) | 2.8 (1–5) |
| # High performers, low participators | 2 | 5 |
| # High performers, high participators | 4 | 4 |
| # Low performers, low participators | 2 | 1 |
| # Low performers, high participators | 1 | 0 |

size allows for the detection of the most salient ideas (rather than saturation of all ideas). They found that sample sizes as low as 10 interviews allowed for researchers to discover the most salient ideas. Thus, we concluded that our sample size of 19 interviews was sufficient.

## Interview protocol

The undergraduate interviewer followed the guidelines and training in Kvale and Brinkmann's *Learning the Craft of Qualitative Research Interviewing*, including suggested exercises and practice scenarios [24]. The undergraduate interviewer then conducted seven pilot interviews that were recorded. These pilot interviews were each reviewed with the principal investigator (EGB) so that mistakes could be corrected, and the interviewer would make changes before the next pilot interview.

We used semi-structured interview methodology to allow the interviewer to ask the subjects to expand on their ideas. Example initial questions (see S1 Appendix) were inspired by prior qualitative studies and were aimed at understanding undergraduate student's belonging, performance, and participation [25, 26]. The analysis for this paper focused only on discussions about participation in class, but sometimes students talked about participation when asked about belonging or performance. Therefore, all interview questions are included in the S1 Appendix, but answers to all of these questions were not necessarily analyzed for this study. Due to this semi-structured framework, the interviewer had the ability to ask follow-up questions and change the order of the questions. Thus, not all participants were asked the exact same questions as each interview was tailored to understand each individual student.

Participants were given minimal information about the purpose of the study, because we were interested to see what themes came up naturally. They were told we were interested in class participation, performance and belonging in general but not that we were interested in gendered differences in these areas. The participants did not know the interviewer prior to the interview, although the participants learned general information about the interviewer as they asked questions. Some learned more about the interviewer (e.g., age and major of the interviewer) than others. If participants asked specific questions about the study, the interviewer told them that she could answer them at the end.

Interviews ranged between 40–50 minutes and were audio recorded. Most of the interviews were conducted online using Zoom, but eight of the women were interviewed in person. The interviewer was in a private room for all of these interviews. After each interview the interviewer took brief notes. These notes included her overall impression of the participant and notes on the atmosphere of the interview. Participants were not given the chance to comment on the transcriptions after they were complete.

## Qualitative analysis

We analyzed the interview data using an iterative, inductive thematic analysis approach that pulled largely from thematic network methods [27]. This approach was data-driven and encouraged us to extract meaning from the interview data rather than starting with *a priori* ideas. First, researchers (including authors ES, JS, AH, and CHH as well as two other undergraduate students) read each interview several times to individually create an initial list of repeating ideas. The research team (authors ES, AH, CHH, and EGB) then met together to turn these ideas into a coding framework that included external factors mentioned by students, participation behaviors, and internal emotions and personality traits. Pairs of researchers (ES and CHH, ES and JS) then re-read all the transcripts and coded them independently line-by-line using our coding framework. These pairs then compared their coding. When there was disagreement, coders discussed until they came to a consensus. Next, we (authors ES, JS, and

EGB) pulled groups of quotes that had been coded similarly to read together and queried pairs of codes together (often internal emotions and personality traits with different external factors) to look for themes in similar passages. Next, we talked as a group to refine our themes, connect them in a thematic network, and decide on global themes. Finally, we began writing about our themes, sending us back to the data once more to ensure our discussion of the themes was accurately representing the text. In this process, we refined our global and subthemes once more (to those described in the Results) and chose quotes that provided evidence for the themes.

## Results

Three major themes emerged from our iterative thematic analysis of the interviews. First, there are many external factors that individually impact students' decisions to verbally participate in class, and some encourage participation while others discourage it. Second, each student weighs various factors in complex ways when deciding if they will speak up, and this cognitive calculus can vary by student. Finally, we found that men and women generally talked about similar factors when discussing their participation, but women seemed to experience more fear of peer judgment and men talked more about course material.

### Theme 1: Many external factors impact students' participation

Students spoke of three main categories of external factors that influence their decisions to speak up in class or not: the professor, the course material, and peers. These main themes and their sub-themes are summarized in Fig 1, with green indicating that a factor increases participation and purple indicating that a factor decreases participation. Each of these three main categories of external factors are discussed below. Although some of our planned potential follow-up questions (see S1 Appendix) specifically referred to professors and peers, all participants who mentioned professors and peers brought them up on their own without prompting except one woman participant who was specifically asked about professors without her mentioning them on her own.

### Professor

Eighteen out of 19 participants talked about professors when asked about patterns in their participation and what influences it. This included discussions of the professor's actions and decisions that shaped the class, the instructor's characteristics and personality, and the relationship the student had with the instructor (see Fig 1).

*Professor's actions.* Many of our participants attributed a large portion of their participation as dependent on the professor's conduct and classroom decisions. For example, students mentioned that memorizing students' names, asking open-ended questions, using online discussion boards, and asking students to discuss with a neighbor generally increased their participation, while random call, "calling students out," teasing students, and having a participation quota that must be reached generally decreased students' motivation to participate. Macey said, "Well a big [reason I do not participate] is if I think the professor is going to call me out for asking a dumb question. . . Even the nicest professors will sometimes call you out for that."

Eight of 19 participants also broadly talked about how professors' pedagogical choices influence their participation. All of these students generally said that they, unsurprisingly, participate more in active classes where professors make space for participation by asking questions or leading discussions compared to classes where instructors simply lecture and are just trying to "get through the material." For example, Paul said, "I think a large part of it is. . . what method the professor uses in teaching and whether he allows that time for discussion." Julia

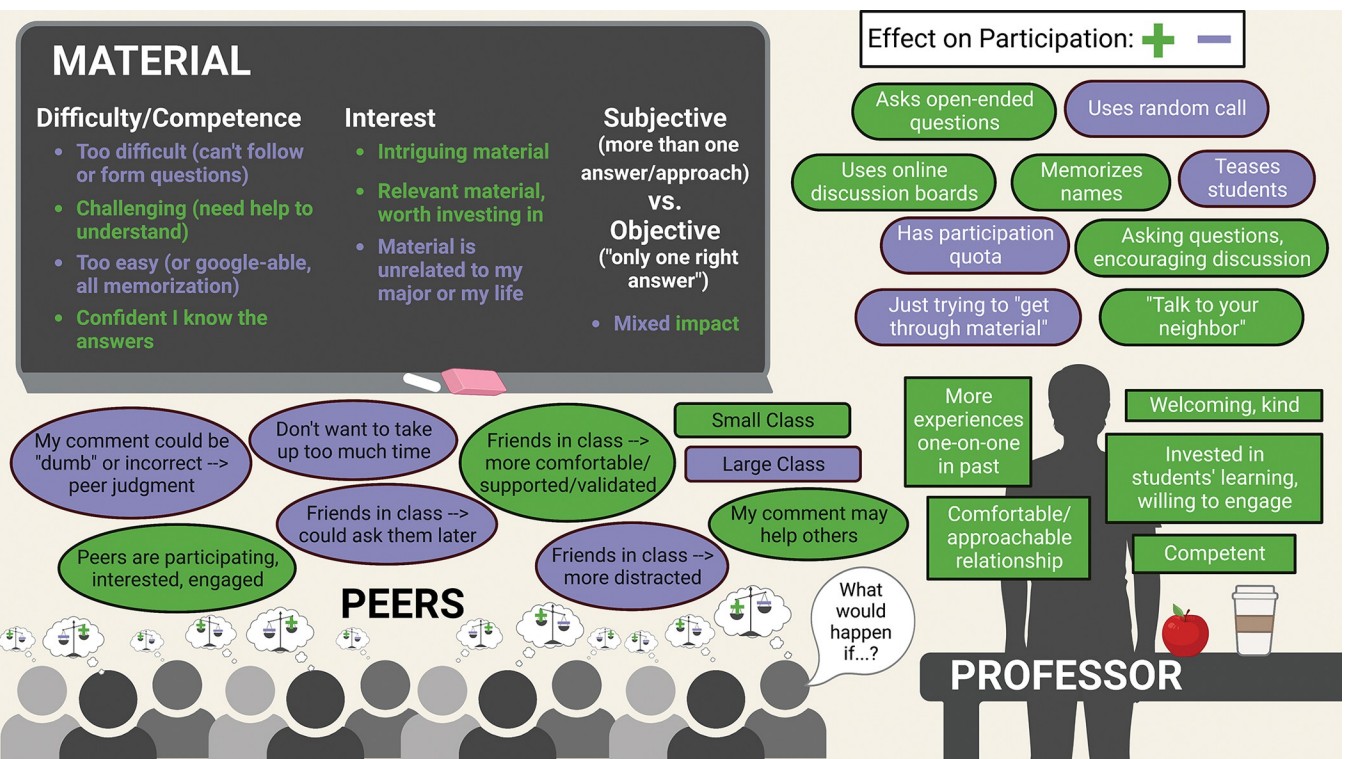

**Fig 1. Summary of main themes from student interviews about what impacts their in-class participation.** Three main categories of external factors are shown: professor, peers, and material. Green suggests a factor increases student participation, and purple suggests a factor decreases student participation. In the professor section, rounded boxes suggest a more concrete action by the professor while the sharp cornered boxes show more subjective professor and relationship characteristics. Scales in students' minds suggest a complex weighing of multiple external factors, which is unique for each student, as they decide whether to participate out loud like the student on the far right. Created with BioRender.com. Printed with permission from BioRender.com, copyright 2023.

also spoke about this, but talked more about how the instructor posed questions and made the class more student-centered:

> That professor was really good at asking questions. And when students ask questions, he would kinda turn it around and have the rest of the class answer and discuss more. . . I think that's actually probably the first time I ever raised my hand in a [computer science] class and voluntarily gave answers and ideas. . . like 'yes, but let's build on that some more' type answers. And that was a first for me.

*Professor's characteristics and relationship.* Participants also spoke about their perceptions of professors' personality traits and attributes. Sometimes the students connected these directly to the concrete actions described above, and sometimes they just spoke of characteristics as something they vaguely discerned. Positive personality traits that encouraged participation were often described as welcoming, making people feel valued, kind, and "how open they seem to questions" (Seth). Anna said that an instructor being "open" made her feel more open and the classroom seem more inviting. Similarly, Shannon said, "if [the professor] seems happy that people are participating. . . it makes me participate more." Leila spoke of her computer science professor when she said, "he was just really kind and really understanding and that's something I really value as a woman in [computer science]. . . He definitely helped me succeed. . . He trusted me." Isabella perceived positive characteristics about her instructor because he memorized their names:

*I just felt like he valued the students and our opinions and stuff, even like those of us that sat in the back and didn't raise our hands very much. He still knew who we were. . . I really appreciate that.*

Similarly, participants described being more likely to participate in classes with professors who are invested in their learning and willing to engage with them. For example, Graham said his professor "asked us engaging questions. . . he really helped me to participate and learn. . . I felt like I was really learning." Kevin spoke of a professor who gave lots of feedback and checked in on students one-on-one: "I feel like they're really trying to help me improve. . . I can tell that they actually cared. . .[the students] felt comfortable."

A couple of students referred to their perception of their instructor's competence as being a factor in whether they would ask a question or not. Derek explained that professors that explain things poorly might get a lot of participation, because students will need to ask a lot of questions for clarification. He also thought that professors who explain things well might trigger participation, but it would be more application-type questions from students. Seth explained what a competent instructor means to him and why he is more likely to ask a question to a professor he perceives as competent:

*There are some professors that I know if I ask a question that they'll be very clear, you know, 'Oh, that's a great question, here's why.' And they'll give a clear explanation, and they'll answer my question. And if I don't need to know it, they'll say, 'don't worry about that. That is a good question beyond the scope of this course, if you're interested, we can talk later.' And so, I know if I ask a question that I'll get something that is relevant, that is concise, and that it answers what I'm saying. There are other professors who I think are great, but if I ask a question, you know, that might lead them on a tangent for five minutes and then they don't answer the question and we just—the whole class just lost five minutes, and so that'll make me less likely to ask questions. Or, you know, sometimes they don't really answer your question or don't understand as well.*

Finally, participants sometimes expressed being more likely to participate if they have positive relationships with professors. This relationship could be built on one-on-one experiences with the professor, such as with Isabella:

*If I have had him or her before or like I've had experiences where I run into them or like I have an opportunity to go and chat with them before the semester starts or at the beginning so that I can have a relationship with them. And then I feel more comfortable participating because I feel like the professor can kind of vibe with what [sic] I'm coming from.*

Even without a lot of one-one-one experiences, participants spoke of approachable professors or instructors that made them feel comfortable being easier to have a relationship with, while students who felt their professors were intimidating felt more space between themselves and the professor. This lack of a relationship seemed to hinder a student choosing to participate. Leila described professors on both ends of the spectrum:

*He was always just really kind, and really I could tell he was really willing to, like, talk to students when they had problems. He wasn't, like, prideful. . . Some professors are like, 'I'm the professor. I'm in charge, like, everything I say is right.' He was, like, always super super chill and like willing to talk about stuff with us and look at issues from lots of different angles. And so, I was really comfortable asking him for help.*

## Material

Seventeen out of 19 participants brought up course material when discussing motivation for their in-class participation. We noticed two main ideas that commonly accompanied students' mention of course material (see Fig 1): the difficulty of the course material (or their personal understanding of and competence regarding the material) and their interest in the material (or the perceived relevance of the material). A few students also commented on whether they perceived the field and/or the professor's questions to be objective versus subjective and how this impacted their participation in class.

*Material difficulty and competence.* First, participants discussed the difficulty of the course material and their perceived competence. About half of the interviewees stated that they are more likely to participate in classes that were more difficult and that they feel little motivation to be active if course material is too easy or dependent solely on memorization. This implies that one goal of participation is to get help with difficult material. About a third of the participants stated this explicitly, that they participate in class to learn or check their understanding. For example, Melanie said that she is more likely to ask questions in classes that require application:

> *[When] it's much easier to understand things conceptually. . .I don't feel the need to participate. So I get it. But then, like obviously once you make the jump [to application], there's always going to be gaps. And so, in order to fill those gaps, you have to participate more.*

However, this was complex, as some students talked about difficulty as if there was a sweet spot, where they are more likely to participate in difficult classes but not in those that are "too difficult" where they cannot follow along and form intelligent questions. This relates to Julia's comment that she had more to contribute in general education courses compared to her major courses in computer science:

> *They were subjects that I grew up thinking about and talking about like biology or math or English. Those kinds of things. Whereas a lot of the classes for my major I'm still just like, 'Whoa, this is all new information. I have nothing to contribute because it's all new.'*

Thus, even though difficulty was generally spoken of as a motivator for participation, some students also said they did not participate in classes where they felt behind or incompetent. For example, Matt said that "A lot of times in class, I'm like too lost to ask questions a lot." Some even said that they would only answer a professor's question when they felt very confident that they knew the correct answer. One student commented that they did not participate in general education courses because they were not as confident in that context as they were in their major courses.

*Interest in material.* Second, a little less than half of the interviewees discussed their interest in the subject or its relevance to them as important motivators for in-class participation. Many of these said, like Graham, that "especially when a topic interests me. . . I participate more." A student's desire to participate in the class would increase if they were intrigued by the material. Likewise, the perceived relevance of the material in the student's life or career path also caused the student to "invest" in the class by participating. Seth said, "But in general. . . I will ask more questions in classes that are related to what I want to do in the future." Similarly, Macey participated more in a class that was "super practical, and it just felt like we weren't doing things just to do them. We were doing them because these were really valuable skills that we were learning." Along these lines, Seth spoke of participating less in his general education courses due to the lack of relevance these courses had for his future.

*Subjective versus objective material.* Finally, a few participants commented on their perception of subjects and/or professors' questions as being more subjective or objective. Both types motivated these few students to participate in different ways. Two students spoke of fields or questions perceived as more objective (i.e., there is only one "correct answer") encouraging their participation because they could be more confident that they were correct or because they wanted to seek clarity on what that correct answer was. Sandy spoke to this when she said, "when things are a little bit more objective, then. . . I tend more to ask questions because I know that there is like an answer that I'm not understanding yet." As our population came from a religious institution where religion classes are part of the general education requirements, religion courses were brought up by multiple participants. Three out of the four students who brought up religion courses said they were more likely to participate in their religion courses compared to others due to the subjective nature of the discussions in those courses. Students talked about how they liked to add to the discussion because there was no right answer and/or because their comment might help another person. The fourth student said that they participate less in religion classes than others because they felt their comments were not needed in a subjective discussion where there was no correct answer. Finally, Julia commented that in computer science, "there are a lot of different routes you can take to get to the same program that's going to do the same thing. So that was something like I felt like I couldn't be wrong in one aspect." Without the pressure of one specific answer, she felt freer to participate in the discussion.

## Peers

Seventeen out of 19 participants talked about their peers when reflecting on their participation habits and causes, so the influence of other students in the classroom was a major theme throughout the interviews for good and for bad. Participants commented on the impact of peers in general, peers with specific characteristics, and the number of peers (see Fig 1).

*Fear of peer judgment.* Seven participants expressed a hesitancy to participate due to a fear of being judged by their peers. These students talked about worrying that they would look dumb if they answered incorrectly, asked a "dumb" question, or made a comment that was not viewed as intelligent. For example, Isabella said "it's just intimidating when there's a bunch of people and you don't know a lot of them and you're afraid to be judged for asking a stupid question." Interestingly, when we probed, participants mostly recounted instances when they observed others becoming embarrassed after asking a "dumb" question, but none of our participants described personally experiencing that feeling of embarrassment from asking a dumb question in class.

Participants' discussion of this fear was often accompanied by a description of classes where they felt behind or less competent. Shannon said, "I was way too scared to raise my hand, because. . . probably everybody knows the answer to this. It's probably a dumb question. So I think I let a lot of things that confused me just like slide by." Macey spoke to the high stakes in her mind when she said, "everybody's super smart so it feels like if I say something dumb it's like, it's over. . . When you think that people might think poorly of you, it just like affects your own psyche." Two women spoke of this in the context of being in the minority as women in a class full of men, and two participants spoke of elevated fear of judgment when they perceived their peers to be more competitive.

*You don't want to look bad in front of your peers and you don't want to. . . appear lesser. [University] is a very competitive environment. . . there is some competitiveness in classes when people are trying to show that they know more or less [than] some people. -Dave*

*Friends as peers*. Eleven of the 19 participants brought up the idea that having friends that they trust or feel comfortable with in their classes affects their participation habits. Most of them said that having a friend in the class makes them more likely to participate. For example, Julia described sitting next to a peer she could relate to in her programming class, and he ended up becoming a good friend. She worked on problems with him and got to know the professor better because of him. This friend naturally broke down participation barriers for Julia and helped her become more comfortable raising her hand and asking questions in class. Some interviewees specifically talked about how having such friends in the class decreased the fear of judgment described above.

On the other hand, one participant said that they actually feel more self-conscious about looking dumb in front of friends and so participate less when friends are in the class. A few other participants also said they are less likely to participate in a class with friends, because they are more likely to get distracted or because they can just ask their friend questions later instead of doing it in front of the class.

*Consideration for peers*. Nine participants mentioned that they made choices about participating out of consideration for their fellow students. Seven of these spoke of wanting to be mindful of others by not taking up valuable class time, either because their questions were too basic for other students or merely due to the amount of time. As Dave said, they "don't want to be that kid that is like always asking questions" and annoys everyone. A few spoke of the content of their questions and comments potentially being beneficial for others, that they could say something that could help someone else or lighten the mood of a stressed classroom.

*Peer group mentality*. About a third of the interviewees talked about a group effect where their peers' culture of participation impacted their own participation habits. In other words, if other students in the classroom were more likely to participate, they themselves would be more likely to participate. Or if others showed interest or excitement about the material, they themselves felt more interest and excitement and desire to participate. In this way, these few participants spoke of participation as if it is contagious. Julia said, "I tend to match the people around me in terms of willingness to put myself out there."

*Number of peers and class size*. Finally, the number of peers in the classroom seemed to be important, since 13 out of 19 participants commented on class sizes when talking about their participation. Students almost unanimously agreed that they are more likely to participate in small classes than in large classes. The reasons for this varied, but often aligned with the peer impacts described above.

When speaking of large classes, participants mentioned participating less because they do not want to take up other students' time (especially if they can just google an answer), do not feel a sense of responsibility to participate (because other people will likely comment or ask a question), are skeptical that they can have an impact as an individual in a crowd, fear judgment if they are wrong in front of a large group of people, do not want to attract attention (perhaps due to their personality), lack a sense of community, are more likely to zone out because they feel anonymous, or because large classes tend to encourage instructors to lecture more.

When speaking of small class sizes, participants spoke of being more likely to participate because it is logistically easier to interact with the instructor and classmates, they feel more connected to other discussion participants, they have less fear of messing up, they feel greater responsibility for the discussion, or they feel more comfortable and less overwhelmed (often due to the relationships they have with their peers).

> *[Small classes are] less overwhelming. . . just because it feels a little bit more approachable. . . And I guess, kind of going off the like not very many other people talk, I feel like if I don't talk nobody else is going to.—Leila*

## Theme 2: Students internally weigh many different external factors when deciding whether to participate

While the external factors described above were commonly mentioned as important to students' participation habits, students' decisions to participate were usually not easily attributed to a single factor. Often, participants mentioned at least two or three external factors when explaining why they chose to verbally participate or not in their classes. In addition, students sometimes discussed the interaction between their personality traits, emotions, or other internal factors with the external factors. We noticed that participants seemed to be weighing the impact of various pieces of the classroom atmosphere and their own internal experience when they reflected on why they participated in some classes and not others. This theme is represented in Fig 1 by the scales in each student's thought bubble as they weigh the positive and negative environmental factors.

We first identified this theme of complex weighing of different factors when we noticed that students sometimes started out describing their participation habits as dependent on "fixed" personality traits but then spoke of themselves as more malleable as the interview progressed. We coded these seemingly "fixed" personality traits when students spoke about them in generalities as if something was always true and no specific context was mentioned (e.g., "I'm just not the type of person who does that"). For example, Olivia first describes her participation in terms of a seemingly fixed personality: "I'm just a pretty quiet person in general. I don't like being the center of attention so... It's not my favorite spot to be." But when asked if there were any specific classes that she was more willing to participate in, Olivia discussed specific contexts in which she felt more comfortable speaking up, bringing up the subjective nature of the material and the professor's pedagogical style:

> I think I tend to participate more in like religion classes... and classes that you're not necessarily asking questions but more like sharing experiences or scriptures or something than like a biology class or computer science class...I think just because, I don't know, those classes look a lot more for... verbal participation. A lot of my other classes, the professors are like, 'this is the lecture... Like if you have questions, you can raise your hand but otherwise, we're just going to go through the material.'

As another example, Stanley first said, "I'm the type of guy that [will fill silence and participate every class]." However, when the interviewer asked if it varies by class, Stanley said, "It depends, not necessarily based on the subject, but just on kind of everything. [It] more depends on the people and the professor and the people around me and how much other people are participating as well."

Sometimes, the internal weighing of different factors was just implied by students bringing up a lot of different external factors in a row. For example, Walter said, "If I'm really good at a class, and it's a really small class, and nobody's answering, then I'll step in if the teacher really seems like he wants an answer." In this case, there is no evidence that one of these is most important, but rather there are many conditions that need to be met for Walter to participate. Similarly, Anna mentions factors from all three categories when describing the ideal environment for participation:

> So it just kind of depends on, I guess, my familiarity with the subject. And also, the teacher too can have an impact, where if the teacher is more open, I'll probably be more open because the classroom feels more inviting. Whereas if the professor's more shut off, like 'no that's wrong,' I'll definitely pipe down, because I don't want to be called out in front of everyone.

At other times, they explicitly talked about the weighing of different factors and which one is more important. Below, we share some examples of students weighing the impacts of our three main categories of external factors: Professors, Material, and Peers.

## Number of peers versus other peer factors

First, we saw multiple examples of students weighing the number of peers in their classes with other factors related to peers. As quoted above under Peers, Isabella told us about feeling intimidated to ask a question in front of her classmates because she worried she would be judged for asking a stupid question. However, right after that quote, she said, "but if there's only a few people, like who cares [about being judged]?" Thus, we see her weighing the risk of judgment with the number of people who could be making those judgments. If the number was small enough, the perceived risk was low enough to motivate her to ask her question.

Jeanette first told us that class size was the most important factor she considered when deciding whether to participate, but she then talked about a class where she had an opportunity to get close to her TA and members of a small group. Not only did she feel comfortable in that small group setting, but that then led to her feeling comfortable in front of the large class: "And I was fine! I, like, talked all the time, and I would go up and do problems on the board just 'cause I was comfortable with the TA and the other students." Thus, even though she had stated that the size of the class was the most important factor, her example suggested that her relationships with her peers and the comfort associated with those relationships could outweigh the negatives of a large class.

Finally, Anna talked about weighing not only the size of the class, but also her physical position in the class when considering her participation habits.

*The front row of big classrooms is kind of scary, just 'cause it feels like you're at the front row of a giant crowd. And I don't know, I just feel like if I'm sitting in the middle of the crowd, it's almost more safe 'cause I'm just one of the bunch. So, if I do ask a dumb question, it's like, 'oh, where were they? oh well.' Whereas if you're in the front row, it's like, 'oh you're the guy in the front row who always falls asleep and then is like, what's going on?' So yeah, I guess that pressure just isn't there in smaller classes.*

## Professor factors versus peer factors

We found a few examples of participants weighing professor factors and peer factors, and we noticed that the professor generally had a larger impact on the students' willingness or ability to participate than their peers did when they explicitly spoke of comparing them. For example, Anna explicitly said that her professor's characteristics mattered more to her, but we can also see her weighing the number of peers, how well she knows those peers, and the risk of peer judgment:

*I think if it's, like, a nice professor in a big room versus a scary professor in a little room, I'd be more likely to participate in the big group even though there's more people. Just because, I don't know, I guess a professor holds a little more sway in my opinion. . . If I am in a big room and the professor is nice and I do make a silly mistake, it's kind of like there's a crowd of people who I won't have to see again. Well, I will, but we won't recognize each other, because like in a big lecture hall, you probably will only recognize a few people that you usually sit by. So, it's kind of like, if I do get called out, yeah, it's embarrassing, but it will last a day.*

On the other hand, Macey initially came to the opposite conclusion when weighing the professor and the risk of peer judgment, with peer judgment risk holding more weight. However,

eventually she acknowledged that professors can make pedagogical choices to help her participate (at least in some ways):

> *And some of these classes have really good professors. Like, it's not the professor's fault at all that I'm not participating. And then in some classes, like, I do have bad professors, and I'm like, 'Yeah you are the reason I'm not participating.' . . .But I mean a lot of it is kind of a peer judgment sort of thing. But, like, no matter what the instructor does, like, I'm not going to talk in front of 200 people. . . But if we're, like, doing, if they're like, 'Okay, turn [to] your partner and talk about whatever we just talked about,' then I actually find that, like, really nice, because, like, we have something to talk about.*

Similarly, Graham mentioned that when the professor asks them to talk to their neighbor or work in small groups during class, it "just kind of eliminates that, I guess that fear for people, or especially for me, that you're wrong or that you mess up or that you're not going to say something right." Paul also spoke of instructors' pedagogical choices having a large impact in large classes:

> *I think it's mostly the size of the class and the way it's set up. I've had [computer science] classes that have been in lecture halls with 300 people that have still been set up well to have discussion. I think a large part of it is how the professor, you know, what method the professor uses in teaching and whether he allows that time for discussion.*

Finally, Dave discussed how instructors' pedagogical actions and teaching style create a classroom culture, but he acknowledged that the instructor has to get student buy-in for that culture to result in participation:

> *That environment that [professors] create through their teaching style. . . It's kind of a culture of just like openness. . . The professor has a very, very large role to play in that: in the culture that they create within their class and the environment that's created. And then obviously, the students have to bring the rest with them.*

### Professor factors versus material factors

We found only one explicit example of a student weighing professor factors versus material factors. In an exchange with the interviewer, Kevin first described a professor he really admired who inspired him to participate and engage because he was passionate, made the class interesting, and tried to connect with students. Kevin then compared this professor to others who just lecture:

> *They're just there to teach. I mean, they'll try to be, you know, entertaining and everything, but there's been some lectures that I was like mid-lecture, I just packed up my stuff and I left. [Interviewer: Really?] Yeah. [Interviewer: And that being because of the professor?] Not necessarily because of him or her. It wasn't—I had no reason to be invested. Other than that, I'm taking the class for my credit or just to learn. I have no other reason to be invested there.*

In this instance, Kevin was commenting on things professors can do to encourage participation, but the most important thing to him seemed to be the relevance of the material to his future.

## Material factors versus peer factors

Most of the examples we found of participants explicitly weighing different factors was between the material category and the peer category. Interestingly, students who do not participate very often usually spoke of the peer factors being weighed more heavily, while those to participate often spoke of the material factors winning out.

While many students talked about needing help or having interest in a topic motivating them to participate, they had to weigh this with whether their participation was a good use of their peers' time and whether their peers would judge their participation. For example, Sandy said that even if the material "might be interesting. . . it's, like, not important to the rest of the class per se, and so. . . I don't want to disrupt the class." Thus, she weighs her consideration for her peers' time more heavily than her interest. Graham has a similar perspective: "And when it interests me, I participate more. But when it comes to, like, the big lecture halls, I would say I participate less just because of that fear [of being wrong]." Isabella spoke of only being able to overcome that fear if she is very confident that she understands the material well enough to form a good question:

> On a good day, if I understand what the professor was talking about and I do have a question, I can raise my hand. Like, I'm not afraid of participating if I understand the question I'm asking.

Seth and Walter both said they are confident enough to ask a question if they want to, but they sometimes choose not to do it if it would only benefit them:

> Sometimes I just don't feel like it's worth the effort of taking everyone else's time to ask a question. And so I just figure, I'll ask my friend or I'll look it up later or I'll ask a T.A. It's kind of the balance of, you know, if like this would be beneficial and have to ask or should I just kind of table it and ask it for later if it's not very relevant or if it's something that I feel like only I would be asking.—Seth

> In my [computer science] class I had a lot of programming experience, so I did talk a lot. I was probably annoying, just 'cause I had the answers and nobody had the answers. And I'm like, 'Come on, guys, this isn't that hard.' So that's probably when I answered the most. And still, I try to limit myself so other people wouldn't be annoyed by me and would have a chance to answer and ask questions.—Walter

Shannon had a lengthy description of how she weighs the motivation of staying engaged and learning the material with fears about her peers. In summary, the course material is a huge motivator for her that generally leads to her participating in class a lot, whereas she is very quiet socially. She ended the dialogue with this summary:

> In class. . .I really want to know. Like, I really want to learn. And so, I'm like, I don't care if, like, this question is dumb, or like, I don't really care if people are annoyed that I'm, you know, taking class time to say this, because I'm trying to learn and that feels more important to me. And then in social settings, I think, yeah, I think probably my shyness or self-consciousness can have a stronger effect on me, because like I let it, I guess, have a stronger effect on me, because. . . I don't have as much of a drive to overcome it. But knowledge is enough of a drive for me.

**Theme 3: Women's and men's experiences largely overlapped, but we saw evidence that women put more weight on their peers while men talked more about course material.** So

far, we have discussed the numerous external factors that impact students' participation habits as well as the complex internal calculus that students conduct as they decide whether to participate. As our third major theme, we noticed that although the experiences of men and women mostly overlap, there was a tendency for women to focus more on peer considerations and men on the course material.

Women's greater focus on peers and men's on course material is first seen in discussions of class size. As described above, students who brought up class size and the number of peers present almost universally agreed that smaller classes were easier to participate in. However, this was more common among women: nine of the 10 women mentioned class size compared to four out of nine men. Furthermore, when citing their reasons for preferring smaller classes over larger ones, eight out of those nine women talked about nerves or fear of judgment in large classes compared to comfort in small classes. For example, Olivia said her reason for preferring small classes is that, "I just would know a greater majority of the people and be comfortable with them." The ninth woman said she worried about taking up too much time in a large class. On the other hand, only one man out of the four who talked about class size mentioned having more fear or less comfort in a large group. Instead, all four men talked about participation being logistically easier in small classes, due to the greater ease with which discussions and interactions could take place, compared to large classes which commonly rely on more lecture-based teaching. Paul described large classes as "usually [in] giant lecture halls and a ton of people and not much question and answer." While not directly about class size, comments about office hours also suggested that women had more fear of peers than our male participants. Four of 10 women talked about feeling more comfortable talking to their professors in office hours when peers were absent, while only one man mentioned this idea. For example, Sandy said "I'm not a huge verbal participater in class. I think that's because I'm like still processing everything, and. . . I shy away from it too. So I go to office hours a lot."

Gender differences in views on class size could also be related to majority versus minority status. Three women brought up issues about being one of very few women in the class. One of these women told a story about a specific TA being very condescending to women students, and the other two women specifically said that being in the minority reduced the likelihood they would participate. For example, Macey said "It's more a thing for computer science, because being part of it is, it's 90% male. So that's like, I feel more embarrassed, I guess. You just think about it differently when it's a bunch of guys."

Next, we saw that women spoke of being more influenced by their fear of or comfort with peers than men did. About half of the women explicitly talked about not participating in class out of fear of looking "dumb," while only about a third of men did. The men who brought this up did not want to look dumb and sometimes worried about their pride being hurt, but this did not seem to be as large of a factor in their participation decisions as the women. For four of the women, this fear of looking dumb was connected to feeling that they were not on the "same level" as their peers in the subject matter (see Shannon's quote above, in the section about fear of peer judgment).

We also noticed gender differences in the way participants discussed having friends in class. Having a good friend in class seemed to be much more helpful to women than men, as six of 10 women but only one of nine men said that friends helped them feel more comfortable participating. For example, Jeanette said, "if I know the people really well then I'll participate more." On the other hand, one of 10 women and two of nine men believed that having friends in class decreased their participation because they were more distracted.

While not as obvious as the gender differences regarding peer factors, we noticed that men were more likely to emphasize course material as important to their participation than were women. For example, five of the nine men said that they were motivated to participate as a

mechanism to learn in class or check their understanding, but only two of 10 women mentioned this idea. For example, when asked why he participates, Derek said "when I don't understand something or I want to try and make a connection to something else to see like—kind of to triangulate what we're talking about."

Similarly, five of nine men talked about participating more when they felt really interested in or intrigued by the material that was being discussed in class, while only two of 10 women spoke of this same kind of intrigued feeling being motivating. For example, Walter said he's less likely to participate when "it's not interesting to me. . . so it's a lot easier to zone out." However, the idea of a gender difference in regard to participation's relationship to interest is contradicted a bit, as more than half of the women talked about participating more when material was relevant to their major or future career, while only about a third of men brought this up.

## Discussion

Like previous studies, we found that instructors, peers, and course material all impact students' in-class verbal participation. Similar to other studies about instructor impacts, we found that students are more likely to participate when they perceive the instructor is warm, open, encouraging, and uses students' names [17, 19] as opposed to "calling students out" or responding negatively to students' comments [20]. We also found that instructors can increase participation through their pedagogical choices, by structuring the class to be more student-centered and providing opportunities for participation as other studies have discussed [16–19]. Mustapha, Abd Rahman [19] mentioned that poor teaching skills discouraged students from participating. Depending on what the students meant by poor teaching skills in that study, this could be related to our finding that students valued instructors who could competently answer their questions, increasing their confidence that it would be worth it to ask their question.

Our findings about peer impacts also support previous literature. Like past studies [18, 20], we found that students generally feel more comfortable participating in smaller classes. This is largely due to a profound fear of peer judgment, or fear of negative evaluation, that has been described previously [10, 15, 19, 20]. Like past studies, we also found that women appear to be more impacted by this fear of peer judgment than their male counterparts [10, 15, 20]. Like Fredricks, Hofkens [16], we also found that peers can have positive impacts, such as when the engagement of other students can be contagious and make a student want to participate more. We also describe the impact of peers on students' consideration of time (participating if they think it will help other students and thus be worth everyone's time).

Finally, other studies similarly found that the course material impacts students' decisions to participate. Like Fredricks, Hofkens [16], we noted that students were more likely to participate when the material was relevant to their personal lives or future careers. Many studies have reported that students are more likely to participate when they feel more prepared and confident about the material [10, 18–20]. We add to this by describing how complex this can be, as students will not participate if the material is too easy and they feel too confident. Thus, there is a middle ground that fosters participation where material is challenging but students think it is within their reach. We found mixed results regarding whether open-ended or more close-ended questions are more encouraging of participation. The importance of material characteristics may have been more important to the men in our sample compared to the women, which may be slightly related to the finding of Auster and MacRone [17] when they found that men were more impacted by the types of questions faculty asked.

While the previous studies mentioned above all describe multiple external factors that impact student participation, one strength of our qualitative design and analysis is the in-

depth look at how students weigh these different factors and the context-dependency of that weighing (both within and across research subjects). We found that students see the classroom as a complex ecosystem, and their decision to participate or not depends on how they view the balance between positive and negative factors in each different class, perhaps even on different days. Even those who initially identified as painfully quiet could identify the right combination of classroom characteristics in which they would feel safe contributing.

### Framing our results within Expectancy Value Theory

After finalizing the broad themes found in our interview data, we returned to existing psychological theories. The external factors that participants mentioned and the way that they described their internal weighing of these factors fit well within the framework of the Expectancy Value Theory (EVT) of Motivation [28, 29]. This theory is commonly used to model the factors that influence students' achievement-related choices (in our case, participating in class). The most proximal social cognitive factors that influence these individual decisions are students' expectations for success and subjective task values [29]. Within subjective task values, six factors have been shown to contribute to overall task value: interest/enjoyment value (i.e., intrinsic value), attainment value (i.e., personal importance connected to individual identity), utility value (i.e., fit within future goals and plans), opportunity cost (weighed against other possible actions), emotional or psychological cost (e.g., anxiety and failure costs), and effort cost [29, 30]. Furthermore, the theory posits that within- and between-individual differences in outcomes (in our case, participating in class) are due to different expectations for success and different subjective task values across individuals and across time for a single individual. In addition, different students have different hierarchies of expectations, costs, and values informed by their socialization and previous experiences [29].

In line with EVT, we found that students will participate if they. . .

- value participating

○ e.g., are interested in the material, need help with challenging material or want to solidify understanding, believe the content is relevant to their lives or future, think participating will help others, or it's required for the class and will thus help their grade

- believe they can successfully participate

○ e.g., lack social anxiety or introversion, perceive enough competency with material, the professor provides opportunities, and they think the professor will successfully answer

- and perceive low cost.

○ e.g., getting called out by professor, being judged by peers, taking up time

In terms of our main themes, (1) the individual external factors tell us what kind of things impact the value, self-efficacy, and cost that students perceive; (2) the complex internal calculus with which students weigh the external factors show the within- and between-individual differences in participation outcomes; and (3) the differences we observed between what men and women emphasized could reflect differences in socialization and/or previous experiences. One quote from Seth perfectly demonstrates a student weighing the costs versus benefits:

*The first thing would be: You're asking this question. Will I, and hopefully the class, gain a deeper knowledge of something that is relevant to the curriculum that could help us either perform better on homework or tests or they could help us in the future as we try to apply this knowledge? And the second thing: by beneficial, I think opportunity cost would—am I*

*confident that the professor can answer this concisely in a way where the time that I take to ask and the time that he takes to answer will be beneficial to the class as a whole?*

## Limitations and future research

First, our conclusions are limited by our study sample. Our purpose with this qualitative study was to deeply understand the complex factors that impact 19 students' decisions to participate in class or not. However, we cannot be confident that our findings generalize to the broader population. While we noted that men and women participants had some differences in the common themes they discussed, we did not aim to statistically compare men and women. Furthermore, there were some differences in our pool of men and women participants. Almost all of the women who agreed to participate in our study were high performers, but we had a broader mix of high- and low-performing men. It is possible that this difference could have contributed to differences in our conclusions by gender. Our participants were also recruited from a private, religious institution, and thus the culture that informs these students' experiences may not be shared with other populations. However, most of the topics these participants discussed are common features in college classrooms everywhere, so they are likely broadly applicable. Future studies could take our qualitative themes and survey diverse, large populations of students to determine if our findings are generalizable and to further investigate differences by gender.

Second, we presented the differences between the interviews of men and women as representing differences in the experiences of men and women in the classroom, likely because of differing socialization. However, it is also possible that men and women have similar experiences but do not feel equally comfortable disclosing them, e.g., their fear of peer judgment. Furthermore, it is also possible that the women felt more comfortable opening up about negative experiences because the interviewer was also a woman. A future study with a male interviewer could possibly reveal different information.

Finally, we were not able to allow participants to comment on the interview transcripts or our findings after the interviews were completed. This feedback would have allowed us to verify that we were correctly representing these participants experiences. Thus, this is a limitation in our study.

## Implications for instructors

Our findings suggest that the reasons students choose to participate in class are complex, so instructors should resist the urge to judge actions simply or harshly (e.g., the quiet students are not intelligent or did not prepare for class). While instructors cannot control many factors that impact participation (e.g., class size, whether students have friends in the class, students' past experiences and socialization), many ideas that came up in our interviews are under the instructor's control.

We recommend that instructors create a student-centered environment by providing opportunities for many students to participate. This can be done through whole-class discussions, smaller group discussions or think-pair-share opportunities (which can make participating in a large class less intimidating), using online discussion boards, or providing other ways for students to participate without verbally speaking in front of everyone. By making a class more student-centered, instructors can also gain important insights about how students are understanding material and adjust teaching accordingly. This could help instructors aim for the intermediate space of providing challenging and engaging material without making it so

hard that students are lost or cannot form questions. Instructors can also facilitate more personal and one-on-one opportunities for connecting with students by memorizing and using names, mingling with students during small group work, and making office hours available and less intimidating. We learned that many students seem particularly worried about being "called out" by professors, so instructors can be mindful of how they respond to student questions and patient with repeat questions.

If instructors want to encourage a gender equitable classroom, our data suggest that fear of peer judgment may be the biggest barrier that yields gender gaps in participation. While instructors cannot control how the students interact with and perceive each other, they can make pedagogical choices that help reduce these fears. Instructors can help create a classroom culture of respect and openness. As mentioned above, smaller group discussions or think-pair-share opportunities allow students to participate in smaller groups with less fear of judgment.

## Supporting information

**S1 Appendix. Semi-structured interview questions.**
(DOCX)

## Acknowledgments

The authors acknowledge Amanda Barrett and Colin Smith, two undergraduate researchers who contributed ideas in large group discussions about main themes after reading the interviews.

## Author Contributions

**Conceptualization:** Emilee Severe, Elizabeth G. Bailey.

**Formal analysis:** Emilee Severe, Jack Stalnaker, Anika Hubbard, Courtni H. Hafen, Elizabeth G. Bailey.

**Funding acquisition:** Emilee Severe, Anika Hubbard.

**Investigation:** Emilee Severe, Elizabeth G. Bailey.

**Methodology:** Emilee Severe, Elizabeth G. Bailey.

**Supervision:** Elizabeth G. Bailey.

**Writing – original draft:** Emilee Severe, Jack Stalnaker, Elizabeth G. Bailey.

**Writing – review & editing:** Emilee Severe, Jack Stalnaker, Anika Hubbard, Courtni H. Hafen, Elizabeth G. Bailey.

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
