## [Decision Letter · Decision Letter 0]

11 Oct 2023

PONE-D-23-19417To participate or not to participate? A qualitative investigation of students’ complex motivations for classroom participationPLOS ONE

Dear Dr. Elizabeth G Bailey,

Thank you for submitting your manuscript to PLOS ONE. After careful consideration, we feel that it has merit but does not fully meet PLOS ONE’s publication criteria as it currently stands. Therefore, we invite you to submit a revised version of the manuscript that addresses the points raised during the review process.

Comments appear at the end of this email.

We look forward to receiving your revised manuscript.

Kind regards,

Nkosiyazi Dube, Ph.D

Academic Editor

PLOS ONE

Journal Requirements:

"The authors acknowledge internal funding from Brigham Young University’s College of Life Sciences awarded to authors ES and AH as a College Undergraduate Research Award. We also acknowledge Amanda Barrett and Colin Smith, two undergraduate researchers who contributed ideas in large group discussions about main themes after reading the interviews."

"The authors acknowledge internal funding from Brigham Young University’s College of Life Sciences awarded to authors ES and AH as a College Undergraduate Research Award. The funders did not play a role in the study design, data collection and analysis, decision to publish, or preparation of the manuscript."

5. We note that Figure 1 in your submission contain copyrighted images. All PLOS content is published under the Creative Commons Attribution License (CC BY 4.0), which means that the manuscript, images, and Supporting Information files will be freely available online, and any third party is permitted to access, download, copy, distribute, and use these materials in any way, even commercially, with proper attribution. For more information, see our copyright guidelines: http://journals.plos.org/plosone/s/licenses-and-copyright.

Reviewers' comments:

Reviewer's Responses to Questions

**Comments to the Author**

1. Is the manuscript technically sound, and do the data support the conclusions?

Reviewer #1: Yes

Reviewer #2: Yes

2. Has the statistical analysis been performed appropriately and rigorously? 

Reviewer #1: I Don't Know

Reviewer #2: N/A

3. Have the authors made all data underlying the findings in their manuscript fully available?

Reviewer #1: Yes

Reviewer #2: Yes

4. Is the manuscript presented in an intelligible fashion and written in standard English?

Reviewer #1: Yes

Reviewer #2: Yes

5. Review Comments to the Author

Reviewer #1: The manuscript intends to highlight the reasons on why students participate or do not participate in class. The rationale for conducting the research is well argued. Of cognizance the research was ethical cleared and evidence to this effect has been uploaded.

The layout presentation is neat and easy to read. An added advantage is a logical flow of the presentation, particularly the methods section, findings and the discussion of the results.

This is a significant study presenting experiences of the participants. Its contribution to the existing body of knowledge will not only benefit the instructors' but the students too.

Reviewer #2: I think that your article is interesting. It is relevant to instructors/lecturers keen to promote verbal participation in the classroom. My main concern is how Theme 3 is presented. I think that you need to include verbatim quotes.

6. PLOS authors have the option to publish the peer review history of their article (what does this mean?). If published, this will include your full peer review and any attached files.

Reviewer #1: **Yes: **Thobeka S Nkomo

Reviewer #2: **Yes: **Dr P.A. Gerrand

Snr. lecturer

Social Work Department

University of the Witswatersand

Gauteng

South Africa

---

## [Author Response · Author response to Decision Letter 0]

10 Jan 2024

FROM EDITOR

PLOS asks that, if an individual is named as the point of contact for data access requests, that authors:

-Please clarify the relationship of person(s) listed as data access points of contact to the data underlying your study.

We have revised it as requested. Specifically, we have included the position title of the non-author point of contact and her relationship with the data.

---

## [Editor Report · Decision Letter 1]

12 Jan 2024

To participate or not to participate? A qualitative investigation of students’ complex motivations for verbal classroom participation

PONE-D-23-19417R1

Dear Dr. Bailey _ Elizabeth,

We’re pleased to inform you that your manuscript has been judged scientifically suitable for publication and will be formally accepted for publication once it meets all outstanding technical requirements.

Kind regards,

Nkosiyazi Dube, Ph.D

Academic Editor

PLOS ONE

---

## [Editor Report · Acceptance letter]

29 Jan 2024

PONE-D-23-19417R1 

PLOS ONE

Dear Dr. Bailey, 

I'm pleased to inform you that your manuscript has been deemed suitable for publication in PLOS ONE. Congratulations! Your manuscript is now being handed over to our production team.

Kind regards, 

on behalf of

Dr. Nkosiyazi Dube 

Academic Editor

PLOS ONE